# Epigenetic Connection of the Calcitonin Gene-Related Peptide and Its Potential in Migraine

**DOI:** 10.3390/ijms23116151

**Published:** 2022-05-30

**Authors:** Michal Fila, Anna Sobczuk, Elzbieta Pawlowska, Janusz Blasiak

**Affiliations:** 1Department of Developmental Neurology and Epileptology, Polish Mother’s Memorial Hospital Research Institute, 93-338 Lodz, Poland; michal.fila@iczmp.edu.pl; 2Department of Gynaecology and Obstetrics, Medical University of Lodz, 93-338 Lodz, Poland; anna.sobczuk@umed.lodz.pl; 3Department of Orthodontics, Medical University of Lodz, 92-217 Lodz, Poland; elzbieta.pawlowska@umed.lodz.pl; 4Department of Molecular Genetics, Faculty of Biology and Environmental Protection, University of Lodz, Pomorska 141/143, 90-236 Lodz, Poland

**Keywords:** epigenetics, CGRP, pain transmission, migraine, *CALCA*, DNA methylation, histone modification, miRNA, lncRNA, circRNA

## Abstract

The calcitonin gene-related peptide (CGRP) is implicated in the pathogenesis of several pain-related syndromes, including migraine. Targeting CGRP and its receptor by their antagonists and antibodies was a breakthrough in migraine therapy, but the need to improve efficacy and limit the side effects of these drugs justify further studies on the regulation of CGRP in migraine. The expression of the CGRP encoding gene, *CALCA*, is modulated by epigenetic modifications, including the DNA methylation, histone modification, and effects of micro RNAs (miRNAs), circular RNAs, and long-coding RNAs (lncRNAs). On the other hand, CGRP can change the epigenetic profile of neuronal and glial cells. The promoter of the *CALCA* gene has two CpG islands that may be specifically methylated in migraine patients. DNA methylation and lncRNAs were shown to play a role in the cell-specific alternative splicing of the *CALCA* primary transcript. CGRP may be involved in changes in neural cytoarchitecture that are controlled by histone deacetylase 6 (HDAC6) and can be related to migraine. Inhibition of HDAC6 results in reduced cortical-spreading depression and a blockade of the CGRP receptor. CGRP levels are associated with the expression of several miRNAs in plasma, making them useful peripheral markers of migraine. The fundamental role of CGRP in inflammatory pain transmission may be epigenetically regulated. In conclusion, epigenetic connections of CGRP should be further explored for efficient and safe antimigraine therapy.

## 1. Introduction

Molecular mechanisms of migraine pathogenesis are incompletely known. This is the main reason this disease remains undertreated despite significant therapeutic progress made by the application of drugs based on monoclonal antibodies against the calcitonin gene-related peptide (CGRP) and the antagonists of and antibodies against its receptor in migraine prevention and therapy [1,2]. Introducing these drugs to the market was a breakthrough in migraine prevention and therapy—they display very high efficacy and tolerability, as shown in recent real-world studies [3,4,5]. However, even if other health cost savings may compensate for the high cost of antibodies targeting the CGRP pathway, an increase in their direct costs remains the main drawback of their use [6].

Genetics plays an important role in migraine susceptibility, symptoms, and comorbidity with other conditions, evidenced by studies associating genetic variants and migraine-related phenotypes (reviewed in [7]). However, these studies do not provide information on the mechanisms that may underline the observed genotype-phenotype association. Gene expression is crucial in the pathway from genotype to phenotype, and emerging evidence suggests that its epigenetic regulation, especially by non-coding RNAs, may play an essential role in acquiring and/or maintaining a specific normal or pathological phenotype [8].

In 2013, Eising et al. pointed out epigenetics as a promising avenue in the study of migraine pathogenesis [9]. Some results and speculations on the role of epigenetics in migraine evoked an increasing interest in an “epigenetic diet” for migraine prevention as intake or withdrawal of some dietary compounds is considered a migraine trigger [10]. We showed that such diets in the case of DNA methylation had limited potential in general and in migraine in particular [11]. Epigenetics was shown to play an important role in the persistent and developmental neuropathic pain models [12,13].

In 2011, Park et al. presented results on the epigenetic regulation of CGRP signaling in trigeminal glia, suggesting their potential in migraine pathogenesis [14]. Further works confirmed the role of CGRP in microglial activation and showed that it might mediate nociceptive signaling via the interaction with its receptors and ATP release in the dorsal horn [15,16,17]. Consistently, several studies suggest that CGRP plays an important role in the regulation of glial gene expression [18]. 

In this review, we present and update information on the role of epigenetics in CGRP regulation and their interaction with other proteins and regulatory RNAs in migraine and other pain-related syndromes. We also present some information on the role of epigenetics in CGRP effects in other diseases, including disorders of the cardiovascular system.

## 2. Calcitonin Gene-Related Peptide—The Gene and the Protein

CGRP is encoded by the calcitonin-related polypeptide alpha (*CALCA*) gene, located at 11p15. The gene has 5684 base pairs (GRCh38/hg38) and encodes the hormone calcitonin (CT) and CGRP alpha (CGRP-1) mediated by alternative mRNA processing [19]. There are two isoforms of CGRP: α-CGRP (CGRP-1), encoded by *CALCA*, and β-CGRP (CGRP-2), encoded by the calcitonin-related polypeptide beta (*CALCB*) gene. CGRP-1 is the predominant form of CGRP expressed in trigeminal ganglia and will be further referred to as CGRP unless otherwise stated [20,21].

The regulation of *CALCA* expression occurs mainly at the transcriptional level. The promoter of the *CALCA* gene contains several elements that may be targeted by transcription factors, including the octamer and two cAMP-responsive elements [22] (Figure 1). The *CALCA* expression in neurons, including trigeminal neurons, is assigned to the activation of an 18-bp enhancer found about 1 kb upstream of the transcription start site (TSS) [23]. It is a part of the distal cell-specific HLH (helix–loop–helix) enhancer. The main activator of the enhancer is the heterotrimer of the bHLH-Zip (basic HLH, leucine zipper) upstream regulatory factors (USFs)-1 and -2 and the forkhead box A2 (FOXA2) that can cooperate with other proteins [14].

CGRP exerts biological action through the interaction with its complex heterotrimeric G-protein coupled receptor, composed of the calcitonin-like receptor (CLR), receptor activity-modifying protein 1 (RAMP1), and a small receptor component protein (RCP) [24] (Figure 2). CLR is a series of seven transmembrane proteins. The presence of a helix-like polypeptide contacting TM7 and embedded into the cytoplasm has also been suggested [25]. RAMP1 is required by CLR to build CGRP, and it is the rate-limiting subunit of the receptor for CGRP binding [26]. The CGRP receptor mediates several signaling pathways and the cyclic adenosine monophosphate (cAMP) response; downstream of the G-protein, Gαs is likely the most important signal transduction pathway for CGRP [27]. As mentioned, the CGRP receptor is therapeutically targeted in migraine by its antagonists and antibodies [28].

The transcription of the *CALCA* gene yields CGRP and CT primary transcripts resulting from the use of two distinct polyadenylation sites and different splicing patterns [29] (Figure 3). As firstly demonstrated in rats, the *CALCA* gene has six exons, of which exons 1, 2, 3, and 4 are spliced together to produce CT mRNA and exons 1, 2, 3, 5, and 6 are spliced to yield CGRP-1 mRNA [29]. Therefore, alternative 3′ splice sites are in exons 4 and 5, and alternative polyadenylation sites are located at the ends of exons 4 and 6. The presence of thermodynamically stable RNA stem-loop forms was shown in vitro at the 3′ splice acceptor of exon 4 of the *CALCA* gene transcript [30]. This RNA secondary structure may play a role in splice site selection and is, therefore, important for CGRP production.

In the alternative processing of *CALCA* mRNA, CT mRNA dominates in the thyroid, whereas CGRP mRNA is preferentially expressed in the central nervous system [31]. CT mRNA specifies the CT precursor in which CT is flanked by a 21 aa potent plasma calcium-lowering peptide, katacalcin [32]. In turn, katacalcin is one of the two known forms of the calcitonin carboxyl-terminus peptide, which may occur as CCP-I (katacalcin) and CCP-II due to the alternative splicing of the primary *CALCA* transcript, in which part of exon 4 splices to exon 5 [33].

CGRP is a 37 aa neuropeptide with four structurally distinct domains (D1–D4) (Figure 4). Domain 1 (aa 1–7) forms a loop/ring-like structure with cysteines at positions 2 and 7, forming a disulfide bridge, which is essential for the biological activity of CGRP [34]. D1 is crucial for the activation of the CGRP receptor. D2 (8–18), D3 (19–27), and D4 (28–37) are generally responsible for receptor binding, and D4 directly interacts with the N-terminal fragment of the receptor [35]. CGRP is structurally similar to amylin (AMY) and adrenomedullin 1 and 2 (ADM1 and ADM2). These four molecules are major members of a family of structurally related peptides [36].

Another form of CGRP, CGRP-2, is encoded by the *CALCB* gene, also located on the 11th chromosome but in a distinct site from *CALCA* [37]. Human CGRP-1 and CGRP-2 differ only by three aa, and, in consequence, they share similar biological activities. CGRP-1 is considered the principal form of CGRP found in the central and peripheral nervous systems, while CGRP-2 is found mainly in the enteric nervous system [38]. Recently, the *CALCB* region has been identified as a migraine risk locus in a genome-wide association study (GWAS) with over a hundred thousand migraine cases [39].

The circulating plasma level of CGRP in healthy individuals is low, but CGRP is naturally increased in pregnancy to regulate uteroplacental blood flow and other vascular changes [40,41]. Changes in plasma CGRP levels may occur in various pathological conditions, and CGRP has been shown to be produced in non-neural cells, including lymphocytes, monocytes, adipocytes, and endothelial cells [42,43]. However, neuronal cells are the main source of CGRP [27].

## 3. Calcitonin Gene-Related Peptide in Migraine

The most remarkable evidence of the involvement of CGRP in migraine pathogenesis is its elevated level in physiological fluids in migraine patients during headache attacks and the induction of delayed migraine-like headaches after CGRP infusion (reviewed in [44]). However, despite such a clear correlation between CGRP and migraine, the mechanism beyond the involvement of CGRP in migraine pathogenesis is not completely clear.

CGRP can be found in many areas of the brain, including the thalamus, the amygdala, periaqueductal grey, locus coeruleus, trigeminal nucleus caudalis (TNC), parabrachial nucleus, hypothalamus, the cerebellum as well as the meningeal vasculature [36]. The trigeminovascular system contains trigeminal ganglia, the branches of the trigeminal nerve and cranial blood vessels and is involved in the regulation of cranial blood flow and pain transmission [45,46].

CGRP is stored in large, dense-core vesicles in the sensory nerve terminals [47]. Trigeminal axons release CGRP into blood vessels of the meninges, causing vasodilation and the activation of trigeminal neurons [48] (Figure 5). Other central nervous system (CNS) and peripheral neurons and glial cells, as well as dural mast cells, are targeted by CGRP, triggering a cascade of effects associated with neuroinflammation, allodynia, and other sensory symptoms of migraine [49].

CGRP is the most potent vasodilatory peptide, and its receptors are localized in the regions that are important in migraine pathogenesis [36]. However, as a vasodilatory factor, it is believed to exert beneficial effects in cardiovascular diseases [27]. It is primarily localized to C and Aδ sensory fibers and has been found elevated in the plasma of patients during spontaneous and nitroglycerin (NTG)-induced migraine attacks [47,48]. Although not all reports confirm these findings [49], several studies strongly suggest that CGRP may play a causative role in migraine as its infusion in migraine patients causes migraine-like headaches [48]. This inspired works on this peptide and its receptor as a target for novel migraine therapies. Two classes of drugs blocking CGRP have been elaborated: small molecular weight antagonists of the CGRP receptor (gepants) and monoclonal antibodies against CGRP or its receptor. The latter include erenumab (Aimovig), fremanezumab (Ajovy), and galcanezumab (Emgality) and are a breakthrough in migraine treatment, but further research to evaluate their efficacy, side effects, and lower their cost is still required [50].

The activation of the CGRP receptor mediates neurogenic inflammation and the sensitization of nociceptive neurons [51]. Reactive oxygen species can activate pro-CT expression from the *CALCA* gene in trigeminal glia by a paracrine regulatory mechanism [52]. It was suggested that this effect occurred following cortical-spreading depression (CSD), an electrophysiological correlate with migraine aura, and neurogenic inflammation to increase CGRP nociceptive events in migraine.

In the CNS, CGRP and its receptor are involved in several signaling pathways that may play a role in migraine pathogenesis [13]. The trigeminal ganglion protrudes to the TNC, where second-order neurons send the signals to the posterior thalamic area (PTA) with receptors for CGRP [36]. Several studies with rodents suggest that the PTA can be an integration center for light and pain, directly related to photophobia in migraine [36,53]. Therefore, CGRP may increase the sensitivity of PTA to sensory stimuli. 

CGRP can also act in the peripheral nervous system, where it targets mast cells, blood vessels, glial cells, trigeminal afferents in the meninges and neural cell bodies, and satellite glia in the trigeminal ganglia [44]. Additionally, CGRP can participate in the induction of neurogenic inflammation by activating mast cells to release compounds that sensitize neurons, causing increased vasodilation in the dura [54]. The modulation of neural activity in the meninges results in the peripheral sensitization of nociceptors [44]. In conclusion, central and peripheral actions of CGRP may cooperate in the induction and/or chronification of migraine. 

Apart from migraine and other pain-related neurological diseases, CGRP may be involved in the pathophysiology of several other disorders, including sepsis, hypertension, diabetes, obesity, atherosclerosis, and myocardial infarction (reviewed in [55]). Therefore, such a broad biomedical significance of CGRP makes it a potential therapeutic target in assorted diseases; however, so far, it has been successfully targeted only in migraine. 

## 4. Epigenetic Connections of CGRP

### 4.1. CGRP and DNA Methylation

Human DNA can be methylated in many ways, but DNA signaling methylation, which is a part of epigenetic regulation of gene expression, involves the transfer of a methyl group from S-adenosyl-methionine (SAM) to the carbon-5 of cytosine in DNA, changing it to 5-methyl cytosine (5 mC). This reaction is catalyzed by the DNA methyltransferases DNMT1, DNMT3A, and DNMT3B [56]. In the human genome, DNA methylation is common and involves mainly cytosine within a 5-′CpG-3′(CpG) dinucleotide—it is estimated that more than 99.9% of DNA methylation may be clustered in CpG dinucleotides [57]. In some regions of the human genome, the frequency of CpG dinucleotides is more than ten-fold higher than on average—they are called CpG islands, which often overlap with gene regulatory regions, including promoters, and are usually unmethylated [58]. Methylated DNA can be demethylated passively or actively [59].

Several studies report an association between DNA methylation and migraine occurrence [11]. In the first GWAS, an 11-year retrospective case–control study assessing 485,000 CpG islands, it was shown that the change from episodic to chronic headaches in mixed headache and migraine patients is associated with the DNA methylation of specific regions in the genome [60]. In another GWAS, 63 differentially methylated regions were identified in migraine patients [61]. These regions included regulatory elements of genes whose products are involved in solute transport: solute carrier family 2 member 9 (SLC2A9, SLC38A4, SLC6A5) and cellular homeostasis: diacylglycerol kinase gamma (DGKG), kinesin family member 26A (KIF26A), dedicator of cytokinesis 6 (DOCK6), and complement factor D (CFD) [62].

The ongoing BIOmarkers of MIGrAine based on the stratification of responders to CGRP monoclonal antibodies (BIOMIGA) proof of concept study (NCT04503083) includes the assessment of DNA methylation levels in migraine subjects as compared to non-migraineur controls (https://clinicaltrials.gov/ct2/show/record/NCT04503083, accessed on 10 May 2022). It is expected that the results of the project will be related to the epigenetic regulation of CGRP [63].

An unmethylated CpG island was found at the 5′ end of the *CALCA* gene, spanning to its first exon [20]. In fact, two CpG clusters were found: one in the upstream region between −1.8 and −0.8 and the other around exon 1. These regions were unmethylated, irrespective of gene expression. That study also showed that the pattern of DNA methylation within the gene correlated with its expression. Specifically, a variable DNA methylation pattern was found in intron 2. Therefore, the expression of the *CALCA* gene may be regulated through the targeting of the constitutively unmethylated CpG island and the demethylation of internal regions of the gene.

Two CpG islands in the human *CALCA* gene have been recently found: one in the distal promoter region (−1833 to −891 from TSS, including the 18-bp enhancer) and one encompassing the proximal promoter and exon 1 [12,64]. Both islands are unmethylated in the *CALCA*-expressing human medullary thyroid carcinoma TT cell line but are hypermethylated in the *CALCA* non-expressing non-small-cell lung cancer NCI-H460 cell line. Hypermethylation was stronger in the 5′ upstream region than in the exon. Incubation of Rat2 cells with 5-aza-2-deoxycytidine (Aza-dC), a DNA methylation inhibitor, resulted in a detectable induction of CT mRNA, also observed when primers specific to both CT and CGRP mRNA were used. 

Recently, Rubino et al. analyzed two CpG islands in the *CALCA* gene: one located in the distal promoter between −2762 and −2362 bp and the other situated in the proximal promoter between −1662 and −1028 bp (Figure 6) in DNA isolated from whole-blood samples obtained from individuals with episodic migraine without aura and healthy controls [65]. They found hypomethylation of two CpG dinucleotides within the proximal region located at −1461 and −1415 in migraine patients compared with controls. Of note, the −1415 site is within the CREB1 binding site. CREB1 is a transcription factor that may play a role in migraine pathogenesis [66]. Some correlations between DNA methylation levels and clinical characteristics in migraineurs were found. No difference in DNA methylation levels was observed in the distal region.

Labruijere et al. studied DNA methylation in many migraine-related genes in different tissues of rats to check whether the results of research on DNA methylation in leukocytes can be extended to other cells more related to migraine [67]. They determined the extent of DNA methylation in *CALCA*, *RAMP1*, CGRP receptor component protein (*CRCP*), calcitonin receptor-like receptor (CALCRL), upstream stimulating factor 2 (*USF2*), and other genes in leukocytes and migraine-related tissues, including dura mater, trigeminal ganglion and trigeminal caudal nucleus. In addition, results obtained in the leukocytes of rats were compared with those in humans. There was no correlation between DNA methylation in rat leukocytes and samples from other tissues. On the other hand, DNA methylation patterns in human lymphocytes correlated with those of rats. This important work suggests two essential aspects of migraine-related research: (1) studies on peripheral tissue may not reflect effects occurring in target tissues, and (2) rats may be a good model to study the significance of DNA methylation in migraine in humans. 

Infusion of CGRP in migraineurs provoked delayed migraine-like headaches, but such an effect was not observed in non-migraineurs [51]. No differences between the methylation of 13 CpG dinucleotides in the promoter of the *RAMP1* gene in leukocytes of migraineurs and controls were shown, but a low DNA methylation trend was observed in migraineurs [68]. However, stratification analysis showed that the DNA methylation level at certain CpG sites was higher in migraineurs with family histories than in patients without such histories. Moreover, methylation of certain CpG sites was lower in female migraineurs than in healthy females. Interestingly, sex-specific differences in RAMP1 mRNA level were observed in a rat migraine model [69]. Therefore, the predominance of migraine in females may be associated with sex-related differences in DNA methylation in genes important in migraine pathogenesis. However, the results of the work of Lubruijere et al., indicating differences in DNA methylation patterns in peripheral leukocytes and migraine target tissues, question these conclusions; in addition, some papers show the utility of DNA methylation peripheral markers in studies on CNS disorders, including Alzheimer’s disease, depression, and bipolar disorders [70,71,72].

In summary, results on the DNA methylation of genes encoding CGRP and its receptor in the periphery may suggest that this epigenetic modification may play a role in migraine pathogenesis, but no direct evidence has been presented so far. It is too early to conclude on the significance of DNA methylation patterns in some specific aspects of migraine, including sex difference in its prevalence and associated symptoms such as aura or comorbidity, but ongoing studies on large cohorts may contribute to our knowledge of the role of DNA methylation in migraine. The process of DNA methylation uses folate (fiolacin, vitamin B9), an essential micronutrient that may link DNA methylation, migraine, and a migraine-preventive diet. However, this issue also needs further research to establish potential dietary recommendations [11].

### 4.2. CGRP and Histone Modifications

The ratio between the total length of the human nuclear DNA (ca. 2 m) and the average size of the nucleus in the human cells (6 µm) dictates packaging the human genome into a highly organized structure (chromatin) not only to fit DNA into the nucleus but also to enable the reading of the primary genetic information contained in DNA. Major protein components of chromatin are histones that are subjected to an array of post-translational covalent modifications, mainly in their N-terminal tails, which protrude from histone complexes and are accessible to histone-modifying enzymes. These modifications are an important part of the information carried by a fragment of DNA associated with histones, and their pattern is referred to as the histone code [73]. It is a combination of covalent chemical modifications to histones, including methylation, acetylation, phosphorylation, ubiquitination, and others, and is established post-translationally by the histone-modifying enzymes that form a large protein complex with DNA-binding proteins and chromatin-remodeling enzymes [74]. These complexes regulate DNA replication, transcription, and repair, as well as other cellular processes in an epigenetic fashion, i.e., independent of DNA sequence [75]. Histone modifications primarily influence the expression of genes, whose promotors are in the regions of chromatin that include such modifications. However, the histone code is not fully known and has a dynamic nature; the number of possible combinations of chemical modifications of all histones is enormously high, and histone modifications are combined with the DNA methylation/demethylation status and action of non-coding RNAs (next section) to produce the epigenetic profile of a cell. 

Histone deacetylase 6 (HDAC6) deacetylases not only histones but also non-histone proteins, including proteins important for steady and dynamic cellular structures [76]. One of the HDAC6 main targets to acetylate in the cytosol is α-tubulin, which creates a heterodimer with β-tubulin to form microtubules to support the cytoskeleton and regulate intracellular transport and cell morphology [77,78]. Bertels et al. aimed to determine if migraine chronification could be facilitated by changes in neuronal cytoarchitecture in an NTG-induced mouse model of migraine and if these changes might be modified by HDAC6 inhibition [79]. They found that chronic exposure to NTG was associated with cytoarchitectural changes in key pain-processing regions. Inhibition of HDAC6 increased the complexity of neuronal cytoarchitecture, which was associated with an increase in the acetylation of α-tubulin. The authors hypothesized that migraine-related cephalic pain might be relieved by restoring migraine-compromised cytoarchitectural complexity. Inhibition of HDAC6 resulted in reduced CSD events, and, finally, blockade of the CGRP receptor reversed NTG-induced chronic allodynia and changes in the cytoarchitecture. Therefore, CGRP may be involved in epigenetically regulated changes in neural cytoarchitecture that may be related to migraine. This involvement concerns both NTG-induced migraine and CSD. This important paper underlines not only the importance of changes in neuronal cytoarchitecture in migraine pathogenesis, including migraine aura, but also points at the disturbed structure of microtubulin, which plays an important role in neuronal communication. Anatomically, these changes are associated with the attenuation of neurite outgrowth and branching. These studies confirmed the importance of epigenetic regulation in the brain structure related to pain processing with the involvement of CGRP. Therefore, recovered neuronal complexity may be a marker of migraine medication with drugs targeting CGRP and/or its receptor.

Abnormal histone modifications, including the trimethylation of histone H3 at lysine 27 (H3K27me3), are associated with the expression of proinflammatory mediators in neuroinflammation, which plays a role in migraine pathogenesis [80,81]. Such modifications can silence gene expression [82]. It was shown that the enhancer of the zeste 2 polycomb repressive complex 2 subunit (EZH2), a histone methyltransferase, facilitated the production of inflammatory mediators in neuropathic pain [83]. Although neuropathic pain is not directly related to migraine, inflammatory mediators can play a role in its pathogenesis. Overexpression of the EZH2 gene and increased H3K27me3 levels in rat spinal microglia was observed [84]. Treatment with CGRP caused the upregulation of H3K27me3 in the spinal dorsal horn and cultured microglial cells through the induction of EZH2. Chromatin immunoprecipitation ChIP-seq data revealed that CGRP changed H3K27me3 enrichments in the promoters of genes that were involved in the regulation of cell growth, phagosome, and inflammation. Therefore, CGRP is involved in neurogenic pain transmission through the methylation of H3 histone mediated by EZH2 in the spinal dorsal horn. Earlier, it was shown that mouse microglial cells (BV2) expressed CGRP receptors and the treatment of microglia with CGRP altered HDAC2 enrichment in the promoters of over 1200 genes in microglial cells; most of the HDAC2-enriched genes were linked to immune- and inflammation-related pathways [18]. Therefore, CGRP may induce an inflammatory response through the induction of differential HDAC2 enrichment in microglia cells. 

The use of the ChIP assay, PCR amplification of the 18-bp enhancer in the promoter of the *CALCA* gene and the antibodies against histone H3 acetylated at Lys9 (H3K9ac), resulted in the observation of a robust signal in neuronal-like rat CA77 cells expressing *CALCA* but not in the rat-embryo-derived Rat2 fibroblast cell line that did not express the *CALCA* gene [14]. The signal of the 18-bp enhancer enriched with H3K9ac antibodies in CA77 cells was greater by almost eight-fold than in Rat2 cells. Therefore, the acetylation of H3 at Lys19 may be important in the cell-specific expression of the *CALCA* gene. However, the use of trichostatin A (TCA), a histone deacetylase inhibitor, did not change *CALCA* expression in Rat2 cells. Treatment with another HDAC inhibitor, sodium butyrate, had little or no effect either. Comparing these results with those obtained for Aza-dC, presented in the previous section, it was concluded that DNA methylation and not histone deacetylation was the major factor controlling *CALCA* expression in Rat2 cells. Treatment with Aza-dC increased about 30-fold the CT mRNA in glial cultures, but treatment with TSA singly did not change *CALCA* expression. However, the Aza-dC + TSA combination resulted in a synergistic, about 80-fold, increase in CT and a 3-fold increase in CGRP mRNA. The effect of histone acetylation needed DNA methylation to be manifested. This important study pointed at glial *CALCA* as an important element in migraine pathophysiology and a potential marker in this disease.

Astrocytes are essential for normal neuronal functions, and there is emerging evidence for their role in migraine pathogenesis [85,86]. So far, there has not been any report associating astrocytes, CGRP, epigenetics, and migraine, but it was shown that CGRP induced the acetylation of H3K9 in astrocytes linked with neuroinflammation in rats with neuropathic pain [87]. Intrathecal injection of CGRP increased the number of astrocytes with acetylated H3K9 in the spinal dorsal horn of rats. ChiP-seq analysis showed that CGRP changed H3K9ac enrichment on gene promoters in astroglial cells. In addition, CGRP treatment increased the expression of H3K9s, C-X3-C motif chemokine receptor 1 (*CX3CR1*), and interleukin 1 beta (*IL1B*) genes in the spinal dorsal horn. A CGRP receptor antagonist inhibited thermal and mechanical hyperalgesia in chronic construction injury rats. Many genes were affected by CGRP-induced changes in their epigenetic profile, including genes involved in proliferation, autophagy, and macrophage chemotaxis. In the context of the present review, that work signals a potential of astroglial cells in pain transmission and also in migraine and points to CGRP as an important modulator of this process through the regulation of the epigenetic profile of these cells. 

Several studies not related to migraine or pain transmission have shown the involvement of CGRP in regulating epigenetic profiles in neuronal cells. Intracerebroventricular administration of CGRP disturbed traumatic fear memories in animal models of post-traumatic stress disorder [88]. This effect was associated with the upregulation of neuronal PAS domain protein 4 (NPAS4), phosphorylated histone deacetylase 5 (HDAC5), and protein kinase D (PKD). CGRP inhibited the interaction between an enhancer in the NPAS4 gene and HDAC5 and increased the binding of the enhancer through acetylated H3 histones. Silencing of protein kinase D weakened the CGRP-mediated increased phosphorylation of HDAC5. Therefore, CGRP may epigenetically function in neural cells through the regulation of the acetylation of H3 histones by interaction with phosphorylated HDAC5 mediated by NPAS4 and PKD. Although these results were not directly related to migraine, in their subsequent work, these authors observed that CGRP elicited photophobic behavior, typical for migraine [89].

In summary, CGRP may be involved in epigenetically regulated changes in neural cytoarchitecture that may be related to migraine and HDAC6 may control this process. Changes in histone methylation may result in changes in the production of proinflammatory molecules that may play a role in migraine. CGRP may be involved in neurogenic pain transmission through the methylation of H3 histone mediated by EZH2 in the spinal dorsal horn. Histone modifications associated with CGRP should be considered with changes in DNA methylation profile.

### 4.3. CGRP and Non-Coding RNAs 

Only a small proportion of the human genome encodes polypeptides, and the human transcriptome can be understood in at least two ways: as the total RNA content of the cell or the complete set of mRNAs. In general, the RNA content of the cell can be divided into coding (mRNA, about 4% of total RNA) and non-coding (ncRNA, remaining 96%) RNAs. The latter can be further divided into housekeeping ncRNAs, such as ribosomal RNA (rRNA) and transfer RNA (tRNA), and regulatory ncRNAs, which can be classified into short non-coding RNA (snRNA, fewer than 200 nucleotides in length) and long non-coding RNA (more than 200 nucleotides). This is not a subject of this review; detailed information on the classification and general properties of non-coding RNAs can be found elsewhere, e.g., [6,90,91]. We will limit our considerations to micro RNA (miRNA), circular RNA (circRNA), and long non-coding RNA (lncRNA). This limitation is a consequence of the lack of information on the connection of many other non-coding RNA types with CGRP. Various miRNAs can recognize complementary sequences not only in mRNAs but also in other regulatory RNAs, including circRNA and lncRNAs. Therefore, a single lncRNA or circRNA can bind several different miRNAs, preventing them from binding their targets. This effect is called sponging, as circRNA and lncRNA can act as a “sponge” against miRNAs.

#### 4.3.1. Micro RNAs

MiRNAs are synthesized from a precursor that has a stem-loop structure by its cleavage to give molecules of miRNAs, typically 20–25 nucleotides in length [92]. They are involved in RNA interference (RNAi), resulting in the silencing of mRNAs, usually occurring due to a miRNA base-pairing to a region in the target mRNA, creating a double-stranded structure that is degraded by a nuclease essential for RNAi. 

Higher levels of CGRP, miR-382-5p, and miR-34a-5p in the plasma of chronic migraine (CM) patients who overused medications compared with patients with episodic migraine (EM) were observed [93]. After adjusting for age, sex, and disease duration, the expression of these miRNAs was still higher in CM than EM patients, while CGRP plasma levels were not correlated with migraine phenotype, although they were positively correlated with the expression of both miRNAs. Therefore, expression of some miRNAs correlated with CGRP levels in blood may be a peripheral marker for migraine, stronger than CGRP. These studies were confirmed and extended in a clinical trial, showing a decreased expression of miR-382-5p and miR-34a-5p in migraine-preventive treatment with erenumab [94].

A lower serum level of miR-30a in migraine patients than in non-migraine controls, with increased methylation in the promoter of the miR-30 gene, was reported [95]. Furthermore, the expression of miR-30a decreased in patients with bilateral seizures, persistent pain, and a high pain index. Bioinformatic analysis showed that the *CALCA* gene is the target for miR-30a, and the overexpression of miR-30a caused the downregulation of *CALCA*, while the knockdown of miR-30 resulted in *CALCA* upregulation. A mechanistic study revealed that miR-30a might bind *CALCA* in its 3′-UTR to degrade it. In conclusion, miR-30a may act protectively against migraine to inhibit its progression through a mechanism involving *CALCA* silencing and a decrease in CGRP levels.

An increased expression of miR-34a-5p in the serum of migraine patients during the disease attack but not in the pain-free period was observed [96]. Then, an increase in miR-34a and a decrease in sirtuin 1 (SIRT1) levels in the spinal cord of mice with complete Freund’s adjuvant (CFA)-induced inflammatory pain was reported [97]. Therefore, the involvement of miR-34a in inflammatory pain may be underlined by the negative regulation of SIRT1. However, it was shown that miR-34a-5p upregulated the IL1B/cytochrome c oxidase 2 (COX2)/prostaglandin E2 (PGE2) inflammation signaling pathway, induced apoptosis, and increased the release of CGRP mediated by the inhibition of SIRT1 in rat primary trigeminal ganglion neurons [98]. Therefore, miR-34a-5p may be a therapeutic target in migraine. In similar research, Dong et al. induced orofacial inflammatory pain by complete Freund’s adjuvant [99]. They observed a downregulation of miR-125a-3p and an upregulation of p38 mitogen-activated protein kinase (MAPK) alpha and CGRP in rat ipsilateral trigeminal ganglions after CFA injection compared with control. This study showed that miR-125a-3p might regulate CGRP in inflammatory pain.

The role of miR-155-5p in TNC in chronic migraine using an NTG-induced chronic migraine mouse model was determined [100]. NTG caused hyperalgesia, upregulated CGRP, fos proto-oncogene, and AP-1 transcription factor subunit (c-FOS), increased the level of miR-155-5p, and decreased the mRNA and protein levels of SIRT1. Increased levels of CGRP and c-FOS induced by NTG were abolished by the miR-155-5p antagomir or SRT1720, a SIRT1 selective synthetic activator, which also downregulated MAPK1 and p-CREB, alleviating microglial activation and decreasing inflammatory mediators, tumor necrosis factor alpha (TNF-α), and myeloperoxidase (MPO). On the other hand, the miR-155-5p agomir or EX527, a SIRT1 inhibitor, aggravated neuroinflammation and central sensitization and increased CGRP expression. This work disclosed key functions of miR-155-5p in TNC in the NTG-induced chronic migraine mouse model and underlined the importance of CGRP in these functions. Specifically, CGRP may act together with miR-155-5p in inducing microglial activation and central sensitization via the downregulation of SIRT1, contributing to the pathogenesis of chronic migraine.

CGRP was reported to interfere with the cellular epigenetic profile in effects not directly related to migraine. Li et al. showed that CGRP inhibited apoptosis and ROS production induced by isoprenaline in cultured rat cardiomyocytes through the regulation of miRNA-1 and miRNA-133a [101]. The effect of CGRP was reversed by its receptor antagonist. 

In summary, CGRP levels may positively correlate with the expression of some miRNAs, including miR-382-5p and miR-34a-5p, that are associated with migraine occurrence and CGRP plasma levels. The expression of some miRNAs correlated with CGRP levels may be a peripheral marker for migraine, stronger than CGRP. The expression of miR-30a is negatively correlated with migraine, and this miRNA may downregulate the *CALCA* gene to decrease CGRP levels. miR-125a-3p and miR-34a-5p may regulate CGRP in inflammatory pain pathways. CGRP may act in concert with miR-155-5p to induce the activation of microglial and central sensitization mediated by the downregulation of SIRT1.

#### 4.3.2. Circular RNAs

CircRNA, due to its structure, with no free ends for exonuclease degradation, is more stable than linear RNA and can act as a sponge for miRNAs [102]. CircRNAs are often classified due to their circular structure and length greater than 100 nucleotides [86]. CircRNAs are mainly generated from precursor mRNA by the splicing of downstream exons to upstream exons in a reverse way (back-splicing) [103,104]. Some variants of such non-canonical mRNA splicing can possibly produce circRNA variants, including exonic circRNAs (ecircRNAs), retained intron circRNAs (EIciRNAs), and intronic circRNAs (ciRNAs) [105]. Apart from miRNA sponging, circRNAs can regulate gene expression in several other ways, including direct interaction with the promotor-RNA polymerase II complex and binding expression regulatory proteins [106]. Some circRNAs can be translated to produce functional proteins.

It was shown that CGRP increased the expression of IL6 mRNA in RAW264.7 macrophages [107]. This effect was associated with changes in the expression of many circRNAs, as evaluated by a microarray assay. Validation of this assay showed that CGRP promoted mmu_circRNA_007893 expression. The effect of CGRP on IL6 induction was attenuated by mmu-miR-485-5p, which had five matching regions with mmu_circRNA_007893. Therefore, mmu_circRNA_007893 might regulate the expression of IL6 mRNA through the sponging of mmu-miR-485-5p. This work confirms the role of CGRP in the inflammatory reaction that may play a role in migraine pathogenesis [108]. Additionally, this work shows some elements of an interplay between the nervous and immune systems and reveals the potential of ncRNA in the regulation of this interplay.

It was shown that CGRP regulated the expression of the FOS-like 2 AP-1 transcription factor subunit (FOSL2) in mouse bone marrow mesenchymal stem cells with the involvement of mmu_circRNA_003795 [109]. FOSL2 is included in several signaling pathways, mainly in the transforming growth factor (TGF)-related pathways [110,111]. TGF-β1 was reported to increase in migraine [112]. Downregulation of mmu_circRNA_003795 increased the expression of mmu_miR-504-3p and was associated with a decreased expression of FOSL2, whereas FOSL2 expression and cell proliferation were decreased. Therefore, CGRP can regulate FOSL2 expression mediated by miR-504-3p sponging through mmu_circRNA_003795.

The role of the interplay between circRNAs and miRNAs in CGRP regulation was confirmed by Wu et al., who observed an aberrant expression of mm9_circ_009056 circRNA in CGRP-induced MC3T3 cells [113]. Increased expression of mm9_circ_009056 in the induced cells was associated with the downregulation of miR-22-3p. Although this study was performed in the context of osteogenesis, the presented mechanism may contribute to the possible regulation of CGRP in migraine.

In summary, CGRP regulation of pathways important in migraine pathogenesis can be assisted by the sponging of related miRNAs by circRNAs. 

#### 4.3.3. Long Non-Coding RNAs

LncRNAs are longer than 200 nucleotides, and they are usually divided into five categories: long intergenic ncRNA (lincRNA), long intronic ncRNA, sense lncRNAs, overlapping sense lncRNA, and bidirectional lncRNA. Some lncRNAs can be further processed to produce small ncRNAs, including miRNAs [114]. The fundamental function of lncRNAs is the regulation of gene expression that can be performed directly or indirectly in cis or trans. lncRNAs may form base pairs with a complementary fragment of mRNA, impeding its translation—the antisense function [115]. lncRNAs recruit and/or guide transcription factors to regulate the transcription of the target gene—the guide function. lncRNAs may serve as a platform for chromatin-remodeling proteins—the scaffold function. lncRNAs may recruit miRNAs or transcription factors and sequester them from their target mRNA or DNA, respectively—the decoy function. The latter also includes sponging effects. Many other mechanisms can be involved in the regulation of gene expression by lncRNAs. 

Xu et al. investigated the effect of the Gm14461 lncRNA on pain transmission in the trigeminal nerve of the mouse trigeminal neuralgia model [116]. They observed that Gm14461 knockdown increased (whereas Gm14461 overexpression decreased) the CGRP protein level in primary mouse trigeminal ganglion neurons. Therefore, Gm14461 promoted pain transmission in the trigeminal neuralgia mouse model with the involvement of CGRP, which may be important in migraine pathogenesis.

Although, generally, migraine is not considered a “classical” neuropathic pain syndrome, it has many features of it [117]. To investigate the mechanism of the CGRP involvement in neuropathic pain, Xiong et al. studied the effect of the uc.48+ lncRNA and uc.48+-targeting siRNA on CGRP release in the spinal cord of diabetic neuropathic pain (DNP) rats [118]. They observed higher expressions of lncRNA uc.48+, CGRP, IL1B, and TNF-α in the neuropathic pain animals compared with the control group. However, injection with siRNA targeting uc.48+ resulted in a decrease in the expression of these molecules compared with non-injected rats. The neuropathic pain group showed increased phosphorylation of p38 and ERK1/2, and uc.48+ siRNA reduced phosphorylation of these molecules. Therefore, lncRNA uc.48+ may play an important role in the transmission of DNP by promoting the release of CGRP in the spinal cord, and siRNA directed against this lncRNA may alleviate the hyperalgesia and allodynia of DNP rats by suppressing the release of CGRP in the spinal cord, the inhibition of phosphorylation of p38 and ERK1/2, and the inhibition of release of IL1B and TNF-α.

Epigenetic regulation of CGRP may play a role in its involvement in the mechanisms underlying cancer-induced pain (CIP). Sun et al. observed the overexpression of lncRNA-NONRATT021203.2 in rats with induced CIP [119]. Furthermore, lncRNA-NONRATT021203.2 targeted C-X-C motif chemokine ligand 9 (CXCL9), increased in CIP rats. CXCL9 was mainly expressed in the CGRP-positive dorsal root ganglion neurons that colocalized with lncRNA-NONRATT021203.2. Therefore, epigenetic regulation of CGRP may play a role in its involvement in CIP. The association of cancer and headache is an emerging, clinically important, and not completely known subject, and including migraine and CGRP in this association may shed some light on it.

As mentioned, CGRP was reported to exert beneficial effects in cardiovascular disorders. It was shown that the regulation of CGRP by lncRNAs might be important in the treatment of acute myocardial infarction [120]. That study showed that hypoxia induced pyroptosis, an inflammatory form of programmed cell death, and the overexpression of the myocardial infarction-associated transcript (MIAT) lncRNA in H9C2 cardiomyoblasts. LncRNA MIAT inhibited CGRP transcription via binding to splicing factor 1 (SF1). As expected, overexpression of SF1 restored CGRP transcription and relieved H9C2 cell pyroptosis. Earlier, it was shown that SF1 promoted the splicing of CGRP pre-mRNA [121]. Therefore, alternative splicing of the *CALCA* gene, a process essential for CGRP production, may be regulated by lncRNA MIAT.

In a similar study, Hu et al. showed that the expression of lncRNA 1700020I14Rik and CGRP was downregulated in myocardial ischemia-reperfusion (I/R) injury and hypoxia/reoxygenation (H/R) injury in a myocardial cell culture model [122]. Overexpression of 1700020I14Rik lncRNA or knockdown of miR-297a increased CGRP protein levels, while targeting 1700020I14Rik lncRNA by short-interfering RNA or overexpression of miR-297a induced a reverse effect. Further research showed that the lncRNA 1700020I14Rik/miR-297a/CGRP axis inhibited myocardial cell apoptosis in myocardial I/R injury. Therefore, that study underlines the importance of epigenetic regulation with lncRNA and miRNA for CGRP regulatory functions in cardiovascular disorders. The action of lncRNAs can be mediated by miRNAs and vice-versa. Interestingly, CGRP upregulation ameliorated myocardial cell injury by overexpression of the lncRNA 1700020I14Rik mediated by miR-297a [122]. This effect was underlined by the suppression of apoptosis by the 1700020I14Rik/miR-297a/CGRP axis. Although this work may seem to be far from migraine pathogenesis, many works, especially those of Borkum, consider migraine as a neuroprotective response against oxidative stress in the brain [123,124,125,126]. Although, in general, migraine is not classified as a typical neurodegenerative disease, several reports associate migraine with injuries in the brain resulting from degenerative processes in neurons and other brain cells [127,128]. Therefore, both lncRNA and miRNA may regulate CGRP to protect against neurodegenerative changes in the brain, which can be related to migraine. Specifically, the 1700020I14Rik/miR-297a/CGRP axis was reported to suppress the apoptosis of myocardial cells, but the suppression of apoptosis is also associated with migraine attacks [122,129]. 

In summary, lncRNAs may regulate alternative splicing of the *CALCA* gene, a process essential for CGRP production. CGRP, together with lncRNAs, may promote pain transmission in trigeminal neuralgia mouse models. lncRNAs may play an important role in the transmission of DNP by promoting the release of CGRP in the spinal cord. We speculate that lncRNA regulation of CGRP in cardiovascular diseases may share mechanism(s) with CGRP involvement in migraine pathogenesis.

A short summary of epigenetic effects related to CGRP and its gene that may be associated with migraine is presented in Table 1.

## 5. Conclusions and Perspectives

Epigenetic regulation of gene expression has an essential impact on the expression of primary genetic information contained in DNA into functional proteins determining the phenotype. CGRP uses alternative RNA processing as a part of a developmental strategy of the brain to direct tissue-specific patterns of polypeptide synthesis [19]. This process can be regulated by DNA methylation of the *CALCA* gene and the involvement of lncRNAs in the alternative splicing of the *CALCA* primary transcript.

CGRP is involved in pain transmission, essential for migraine and other pain-related syndromes, and this involvement can be featured by epigenetic modifications of the *CALCA* gene, including DNA methylation, histone modifications, and regulatory ncRNAs. On the other hand, CGRP may change the cellular epigenetic pattern modifying the expression of its components and its regulatory protein.

Epigenetic modifications of the *CALCA* gene are important for its cell-specific expression not only in neuronal cells but also in cultured glia [17]. This opened a perspective to consider the epigenetic regulation of *CALCA* in glial cells as a potential therapeutic target in migraine. The involvement of glial cells in migraine has been addressed in several studies and hypotheses (e.g., [130,131,132]), but mainly in the context of the role of glial cells in cortical-spreading depression. The action of antagonists of and antibodies against CGRP and its receptor also affects glial cells expressing the *CALCA* gene, but the problem of epigenetic regulation of this gene as a target in migraine prevention and therapy is still a perspective [133].

To position *CALCA* and CGRP in the network of genes involved in migraine pathogenesis is equally important as the sole role of this peptide in migraine. As suggested by Andrew Russo, one way to do so would be a study of the epistasis relationships and epigenetic regulators of CGRP [134]. We have tried to show that epigenetics plays an important, yet poorly explored, role in the regulation of the *CALCA* gene, but, on the other hand, we also tried to show that CGRP may also evoke changes in the epigenetic profile of promoters of genes that are important in migraine pathogenesis.

Several reports show the involvement of CGRP in cardiovascular diseases, underlined by epigenetic modifications of pathways that may be important in migraine pathogenesis. Therefore, checking these pathways in migraine seems to be an immediate perspective.

Although this work focuses on the epigenetic aspects of the CGRP pathway, it is worth mentioning that recently, Zecca et. showed for the first time the suitability of the rs7590387 polymorphism of the RAMP1 (receptor activity-modifying protein 1) gene as a predictive marker of the efficacy of erenumab, the first monoclonal antibody against the CGRP receptor approved for migraine prevention [135]. Earlier, it was shown that DNA methylation at the promoter of the RAMP1 gene might play a role in migraine pathogenesis [68].

Many aspects of CGRP functioning require explanation. The apparently paradoxical observation that salmon CGRP is more efficient in humans than human CGRP is still not completely clear, especially how its administration affects pain perception [136,137]. These problems can be clarified by investigating the epigenetic aspects of CGRP functioning, but, first of all, research on CGRP epigenetic connections should be aimed at the development and improvement of CGRP-based antimigraine therapy.

## Figures and Tables

**Figure 1 ijms-23-06151-f001:**
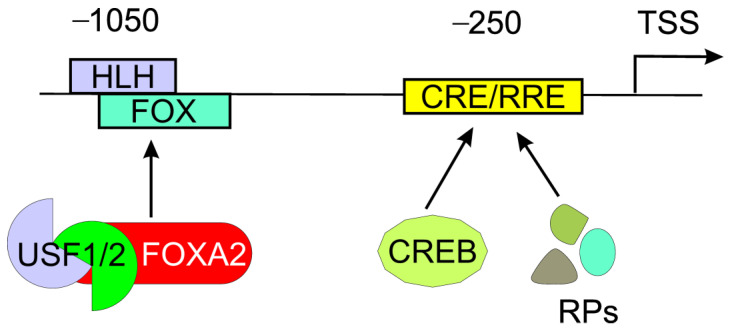
The main regulatory element in the promoter of calcitonin gene-related polypeptide alpha (*CALCA*) is an 18-bp enhancer, which is a part of a distal cell-specific complex helix–loop–helix (HLH) enhancer. The HLH motif of the 18-bp enhancer overlaps with a forkhead (FOX) motif and is targeted by heterotrimer upstream stimulatory factor (USF)-1 and -2 and the cell-specific forkhead box A2 protein (FOXA2). The promoter also contains a proximal cAMP-responsive element (CRE) and a Ras responsive element (RRE) targeted by the cyclic adenosine monophosphate (cAMP) response element-binding protein (CREB) and related proteins (RPs). Approximate positions of these two elements, upstream of the transcription start site (TSS), are indicated. The *CALCA* promoter contains both cell-specific and non-cell-specific elements as well as CpG dinucleotides contributing to functional CpG islands not presented here.

**Figure 2 ijms-23-06151-f002:**
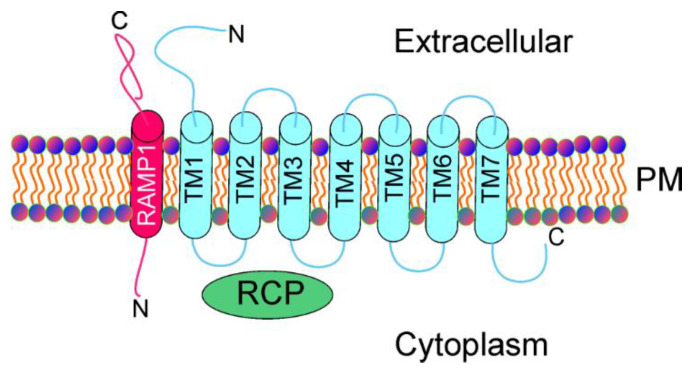
Calcitonin gene-related peptide receptor, a complex heterotrimeric G-protein-coupled receptor, consists of the calcitonin-like receptor (CLR), receptor activity-modifying protein 1 (RAMP1), and a small receptor component protein (RCP). CLR includes 7 transmembrane proteins (TM1–7), whereas RAMP1 is a single transmembrane protein. PM—plasma membrane.

**Figure 3 ijms-23-06151-f003:**
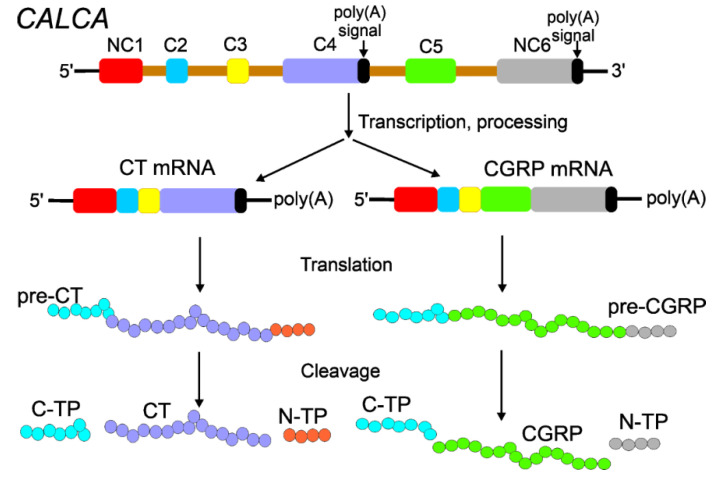
Alternative processing of the *CALCA* gene produces calcitonin (CT) and the calcitonin gene-related peptide (CGRP). The *CALCA* gene has 6 exons separated by 5 introns (gold). Exons 1 and 6 are non-coding exons (NC1, NC6), whereas exons 2–5 are coding exons (C2–C5). Exons 4 and 6 contain signals for polyadenylation (poly(A) signals) that are linked with termination signals in the transcription of the *CALCA* gene. Therefore, two different *CALCA* pre-mRNAs having common NC1 + C2 + C3 regions are produced, bearing polyadenylated (poly(A)) tails at their 3′ ends. These two mRNAs are then spliced to produce CT mRNA with four first exons with a poly(A) tail at the 3′ end of exon 4 and CGRP mRNA with three first exons plus exons 5 and 6 with a poly(A) tail at its 3′ end. These two mRNAs are translated to produce CT and CGRP precursors. Post-translational cleavage results in functional CT and CGRP proteins as well as N- and C-terminal peptides (N-TP and C-TP, respectively).

**Figure 4 ijms-23-06151-f004:**
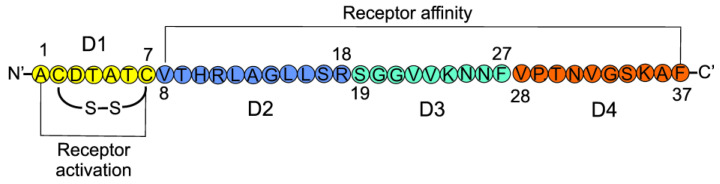
Linear representation of the calcitonin gene-related peptide with its four domains (D1–D4) marked in different colors. Amino acids are denoted according to the one-letter code. Numbers indicate the positions of amino acids from the N-end. The N- and C-terminals are denoted with primed letters to distinguish them from amino acid symbols.

**Figure 5 ijms-23-06151-f005:**
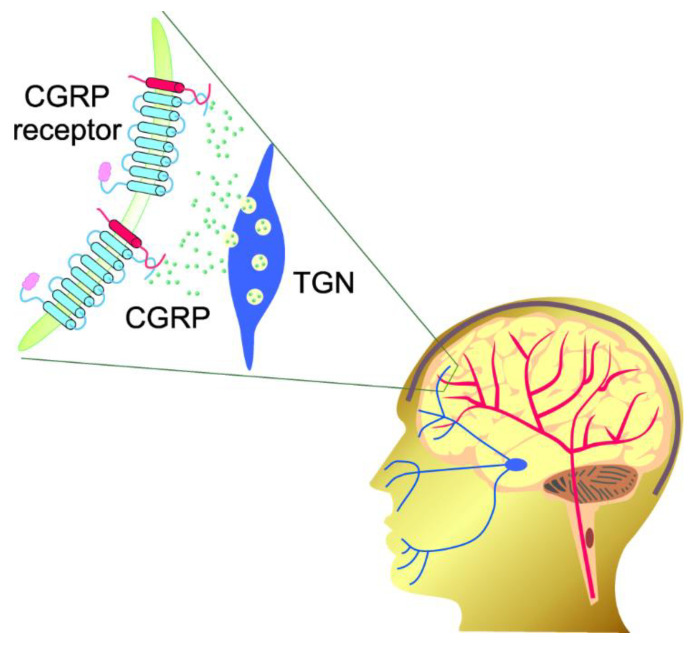
Calcitonin gene-related peptide (CGRP) is stored in large dense-core vesicles in the sensory nerve terminals, but after activation of the trigeminovascular system, CGRP molecules (small green objects) can be released at trigeminal nerve (TGN, blue) endings and bind the CGRP receptors present in both TGN and cerebral blood vessels (red).

**Figure 6 ijms-23-06151-f006:**
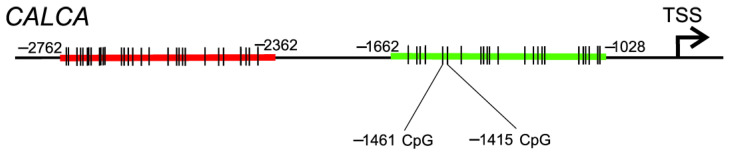
Two CpG islands have been identified in the promoter of the human *CALCA* (calcitonin-related polypeptide alpha) gene: proximally located between −1028 and −1662 from the transcription start site (TSS) (green) and distally placed between −2362 and −2762 (red). The CpG sites within either island are only symbolically marked by short vertical bars. Positions of two CpG sites located at −1461 and −1415 and recently identified as hypomethylated in migraine patients by Rubino et al. [65] are also presented.

**Table 1 ijms-23-06151-t001:** Epigenetic effects related to CGRP ^1^ in migraine and migraine-associated syndromes.

Epigenetic Modification	Effect	Reference
DNA methylation	Hypomethylation of two CpG dinucleotides in the promoter of *CALCA* in sporadic migraine patients	[65]
	Low DNA methylation trend in the promoter of *RAMP1* in migraineurs with higher methylation levels in patients with family history, lower methylation in female migraineurs	[68]
Histone modifications	Blockade of the CGRP receptor by HDAC6 inhibition, resulting in reversing of NTG-induced chronic allodynia in mice	[79]
	Co-expression of CGRP-induced abnormal histone modifications, including H3K27me3, with proinflammatory mediators in the spinal dorsal horn and cultured microglial cells	[80,81,84]
	Altered HDAC2 enrichment in over 1200 genes in microglial cells by CGRP, with most of the genes belonging to immune- and inflammation-related pathways	[18]
	H3K9ac signal in rat cells expressing *CALCA,* but lack of such signals in cells non-expressing *CALCA;* that effect of histone acetylation required DNA methylation to be manifested	[14]
	CGRP-induced acetylation of H3K9 in astrocytes linked with neuroinflammation in rats with neuropathic pain and H3K9ac enrichment on gene promoters in astroglial cells	[87]
Micro RNAs	CGRP plasma levels positively correlated with the expression of miR-382-5p, and miR-34a-5p in plasma of migraineurs	[93]
	Overexpression of miR-30a caused downregulation of *CALCA*, while knockdown of miR-30 resulted in *CALCA* upregulation; miR-30a might bind *CALCA* in its 3′-UTR to degrade it	[95]
	miR-125a-3p regulated CGRP in inflammatory pain;	[99]
	increased levels of CGRP induced by NTG were abolished by miR-155-5p antagomir; miR-155-5p agomir aggravated neuroinflammation and central sensitization and increased CGRP expression	[100]
Circular RNAs	CGRP promoted mmu_circRNA_007893 expression in RAW264.7 macrophages	[107]
	CGRP regulated migraine-related FOSL2 expression mediated by miR-504-3p sponging by mmu_circRNA_003795	[109,112]
	Aberrant expression of mm9_circ_009056 circRNA in CGRP-induced MC3T3 cells	[113]
Long non-coding RNAs	Gm14461 lncRNA knockdown increased, whereas Gm14461 overexpression decreased CGRP levels in primary mouse trigeminal ganglion neurons	[116]
	Overexpression of lncRNA-NONRATT021203.2 in rats with induced CIP; lncRNA-NONRATT021203.2 targeted CXCL9, increased in CIP rats; CXCL9 was expressed in the CGRP-positive dorsal root ganglion neurons that colocalized with lncRNA-NONRATT021203.2	[119]

^1^ Abbreviations: CGRP, calcitonin gene-related peptide; *CALCA*, calcitonin-related peptide alpha; *RAMP1,* receptor activity-modifying protein 1; HDAC6, histone deacetylase 6, NTG, nitroglycerin; H3K27me3, trimethylation of histone H3 at lysine 27; H3K9ac, histone H3 acetylated at lysine 9; FOSL2, FOS-like 2 AP-1 transcription factor subunit; CIP, cancer-induced pain; CXCL9, C-X-C motif chemokine ligand 9.

## Data Availability

Not applicable.

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
