# Peer review of "Epigenetic Connection of the Calcitonin Gene-Related Peptide and Its Potential in Migraine"

_ijms, 2022, doi:10.3390/ijms23116151_

Round 1

Reviewer 1 Report

This review summarizes the available evidence on CGRP epigenetics and migraine.

It is an interesting topic, setting the bases for new therapeutic insights.

I have a few comments:

  • The writing sometimes is hard to read. A revision by a native English speaker is advisable.
  • A figure explaining the involvement of CGRP in migraine would increase the understating of the topic also in non-headache specialists.
  • When discussing the outcomes of monoclonal antibodies targeting the CGRP pathway, please highlight that the main drawback of their use is an increase in direct costs (even if other health cost savings possibly compensate for it. See Mahon et al., 2021). Tolerability and efficacy also in real-life were proved very high (see Vernieri et al, 2021 PMID 33941080; Barbanti et al, 2021 (PMID 34309862); Barbanti et al., 2022 (PMID 35397503 )
  • Please also comment on the first study assessing the genetic predictors of response to monoclonal antibodies, more specifically erenumab (Zecca et al, 2022 PMID: 34965002)

Author Response

This review summarizes the available evidence on CGRP epigenetics and migraine. It is an interesting topic, setting the bases for new therapeutic insights.

I have a few comments:

Comment: The writing sometimes is hard to read. A revision by a native English speaker is advisable.

Answer: We have done our best to improve the style of the manuscript.

Comment: A figure explaining the involvement of CGRP in migraine would increase the understating of the topic also in non-headache specialists.

Answer: We have added a new figure to section 3. Calcitonin gene-related peptide in migraine. It is Figure 5 in the revised version.

Comment: When discussing the outcomes of monoclonal antibodies targeting the CGRP pathway, please highlight that the main drawback of their use is an increase in direct costs (even if other health cost savings possibly compensate for it. See Mahon et al., 2021). Tolerability and efficacy also in real-life were proved very high (see Vernieri et al, 2021 PMID 33941080; Barbanti et al, 2021 (PMID 34309862); Barbanti et al., 2022 (PMID 35397503 ).

Answer: We have replaced the following fragment in Introduction:

“However, it is too early to draw a definite conclusion on the full success of these drugs, especially that their long-term safety has not been confirmed and they have several serious limitations, including high proportion of non-responders, adverse effects and high cost [1].”

with

“Introducing of these drugs to the market was a breakthrough in migraine prevention and therapy – they display very high efficacy and tolerability as it was shown in recent real-world studies [3-5]. However, even if other health cost savings may compensate for high cost of antibodies targeting the CGRP pathway, an increase in their direct costs remains the main drawback of their use [6].”

with new references:

  1. Barbanti, P.; Aurilia, C.; Cevoli, S.; Egeo, G.; Fofi, L.; Messina, R.; Salerno, A.; Torelli, P.; Albanese, M.; Carnevale, A.; et al. Long-term (48 weeks) effectiveness, safety, and tolerability of erenumab in the prevention of high-frequency episodic and chronic migraine in a real world: Results of the EARLY 2 study. Headache 2021, 61, 1351-1363, doi:10.1111/head.14194.
  2. Barbanti, P.; Egeo, G.; Aurilia, C.; d'Onofrio, F.; Albanese, M.; Cetta, I.; Di Fiore, P.; Zucco, M.; Filippi, M.; Bono, F.; et al. Fremanezumab in the prevention of high-frequency episodic and chronic migraine: a 12-week, multicenter, real-life, cohort study (the FRIEND study). The journal of headache and pain 2022, 23, 46, doi:10.1186/s10194-022-01396-x.
  3. Vernieri, F.; Altamura, C.; Brunelli, N.; Costa, C.M.; Aurilia, C.; Egeo, G.; Fofi, L.; Favoni, V.; Pierangeli, G.; Lovati, C.; et al. Galcanezumab for the prevention of high frequency episodic and chronic migraine in real life in Italy: a multicenter prospective cohort study (the GARLIT study). The journal of headache and pain 2021, 22, 35, doi:10.1186/s10194-021-01247-1.
  4. Mahon, R.; Huels, J.; Hacking, V.; Cooney, P.; Danyliv, A.; Vudumula, U.; Vadapalle, S.; Vo, P.; Maniyar, F.H.; Palmer, S.; et al. Economic evaluations in migraine: systematic literature review and a novel approach. J Med Econ 2020, 23, 864-876, doi:10.1080/13696998.2020.1754840.

Comment: Please also comment on the first study assessing the genetic predictors of response to monoclonal antibodies, more specifically erenumab (Zecca et al, 2022 PMID: 34965002).

Answer: We have added the following fragment to Conclusion and Perspective section:

“Although this work focuses on epigenetic aspects of the CGRP pathway, it is worth mentioning that recently Zecca et. showed for the first time a suitability of the rs7590387 polymorphism of the RAMP1 (receptor activity modifying protein 1) gene as a predictive marker of the efficacy of erenumab, the first monoclonal antibody against the CGRP receptor approved for migraine prevention [137]. Earlier, it was shown that DNA methylation at the promoter of the RAMP1 gene might play a role in migraine pathogenesis [138].”

with new references:

  1. Zecca, C.; Cargnin, S.; Schankin, C.; Giannantoni, N.M.; Viana, M.; Maraffi, I.; Riccitelli, G.C.; Sihabdeen, S.; Terrazzino, S.; Gobbi, C. Clinic and genetic predictors in response to erenumab. Eur J Neurol 2022, 29, 1209-1217, doi:10.1111/ene.15236.
  2. Wan, D.; Hou, L.; Zhang, X.; Han, X.; Chen, M.; Tang, W.; Liu, R.; Dong, Z.; Yu, S. DNA methylation of RAMP1 gene in migraine: an exploratory analysis. The journal of headache and pain 2015, 16, 90, doi:10.1186/s10194-015-0576-7.

Reviewer 2 Report

The paper discusses a very new and, in some ways, still unexplored topic of the pathophysiology of migraine and its wide-ranging application in the therapeutic and clinical settings. 

Author Response

Comment: The paper discusses a very new and, in some ways, still unexplored topic of the pathophysiology of migraine and its wide-ranging application in the therapeutic and clinical settings.

Answer: Thank you.

Reviewer 3 Report

 The paper by Michal Fila et al. presents and updates information on the role of epigenetics in CGRP regulation and its interaction with other proteins and regulatory RNAs in migraine and other pain-related syndromes. In addition, the manuscript presents some information on the role of epigenetics in CGRP effects in other diseases, including disorders of the cardiovascular system.

The topic is timely and may attract much attention. However, in its current version, the manuscript has several limitations that should be addressed.

I have some suggestions to improve this paper:

1.     Line 183 „CGRP is the most potent vasodilatory peptide, and its receptors are localized in the regions that are important in migraine pathogenesis [31].” - Please explain in more detail which of these regions.

2.    There are no references in many places to support the facts described, please insert some references.

For example:

Line 178 „Trigeminal axons release CGRP into blood vessels of the meninges causing vasodilation and activation of trigeminal neurons.”

Line 239 „These regions included regulatory elements of genes whose 239

products are involved in solute transport: ....”

Line 249-264 - there are few references in this paragraph.

4.2. Chapter - there are few references in this chapter.

Line 466-470 „MiRNAs are synthesized from a precursor that has a stem-loop structure by its cleavage to give molecules of miRNAs typically 20-25 nucleotides in length…”

Line 504 – 517 „The role of miR-155-5p in TNC in chronic migraine using NTG-induced chronic migraine mouse model was determined [92]. NTG caused hyperalgesia, upregulated CGRP…”

3.     Please present more articles, preferably more than 150 for review articles. I suggest the authors focus their efforts on researching relevant literature: I believe that adding more citations will help to provide a better and more accurate background to this study.

For eaxmple:

https://pubmed.ncbi.nlm.nih.gov/35052756/

https://pubmed.ncbi.nlm.nih.gov/25219387/

https://pubmed.ncbi.nlm.nih.gov/34635045/

4.     It would be worthwhile to make a table of the results so far on the epigenetic aspects of CGRP and migraine.

5.     Formal errors:

·       In many places, the abbreviated form appears in front and the full name is in parentheses (e.g.: calcitonin gene-related peptide (CGRP); calcitonin-like receptor (CLR) etc.) while elsewhere they are reversed. I think the full form should appear first and the abbreviated form should be put in parentheses.  Please, correct these.

For example:

Line 67 CALCA (calcitonin related polypeptide alpha) vs. calcitonin related polypeptide alpha (CALCA)

Line 71 CALCB (calcitonin related polypeptide beta)

Line 77, 92 HLH (helix-loop-helix)

Line 94 FOXA2 (forkhead box A2)

Line 240 SLC2A9 (solute carrier family 2 member 9)

Line 241 DGKG (diacylglycerol kinase gamma)

Line 242 KIF26A (kinesin family member 26A), DOCK6 (dedicator of cytokinesis 6), and CFD (complement factor D)

Line 376 EZH2 (enhancer of zeste 2 polycomb repressive complex 2 subunit)

Line 382, 392 ChIP (chromatin immunoprecipitation)

Line 417 CX3CR1 (C-X3-C Motif Chemokine Receptor 1)

Line 418 IL1B (interleukin 1 beta)

Line 492 SIRT1 (sirtuin 1)

Line 495 COX2 (cytochrome c oxidase 2)

Line 496 PGE2 (prostaglandin E2)

Line 501 MAPK (mitogen-activated protein kinase)

Line 506 c-FOS (fos proto-oncogene)

Line 511 TNF-α (tumor necrosis factor alpha), MPO (myeloperoxidase)

Line 558 TGF (transforming growth factor)

Line 621 SF1 (splicing factor 1)

·       The abbreviated form appears earlier, the full form shows up only later. The full form of the abbreviation must be written at the first mention and the abbreviated form must be enclosed in parentheses. Please correct these.

For example:

Line 89 cAMP – Line 108 cyclic adenosine monophosphate (cAMP)

·       The abbreviation has previously been introduced. After the first appearance, only the abbreviated form should be used.

For example:

Line 68 calcitonin (CT) – Line 135, 137 calcitonin

Line 493 complete Freund's adjuvant (CFA) – Line 500 complete Freund’s adjuvant

·       The abbreviation should appear in parentheses at the first appearance of the full form, not later.

For example:

Line 136 katacalcin – Line 137 CCP-I (katacalcin)

·       The term below is abbreviated twice.

Line 382 and 392 ChIP (chromatin immunoprecipitation)

Line 501 and 509 MAPK (mitogen-activated protein kinase)

·       CALCB is written in italics in Line 71, while not elsewhere (e.g.: Line 153, 158).

·       Line 70 ẞ- CGRP (CGRP-2) vs Line 153 CGRP beta (CGRP-2) – please unify this.

·       The full form of CNS (Line 180) does not appear anywhere in the text.

·       Line 199 double „suggested”

·       There are som typing errors, please check and correct them.

For example:

Line 367 plying

Author Response

The paper by Michal Fila et al. presents and updates information on the role of epigenetics in CGRP regulation and its interaction with other proteins and regulatory RNAs in migraine and other pain-related syndromes. In addition, the manuscript presents some information on the role of epigenetics in CGRP effects in other diseases, including disorders of the cardiovascular system.

The topic is timely and may attract much attention. However, in its current version, the manuscript has several limitations that should be addressed.

I have some suggestions to improve this paper:

Comment: 1.     Line 183 „CGRP is the most potent vasodilatory peptide, and its receptors are localized in the regions that are important in migraine pathogenesis [31].” - Please explain in more detail which of these regions.

Answer: We have added the following sentence to continue:

“These are the thalamus, the amygdala, periaqueductal grey, locus coeruleus, trigeminal nucleus caudalis, parabrachial nucleus, hypothalamus, the cerebellum and the meningeal vasculature.”

Comment: 2.    There are no references in many places to support the facts described, please insert some references.

For example: Line 178 „Trigeminal axons release CGRP into blood vessels of the meninges causing vasodilation and activation of trigeminal neurons.”

Line 239 „These regions included regulatory elements of genes whose 239

products are involved in solute transport: ....”

Line 249-264 - there are few references in this paragraph.

4.2. Chapter - there are few references in this chapter.

Line 466-470 „MiRNAs are synthesized from a precursor that has a stem-loop structure by its cleavage to give molecules of miRNAs typically 20-25 nucleotides in length…”

Line 504 – 517 „The role of miR-155-5p in TNC in chronic migraine using NTG-induced chronic migraine mouse model was determined [92]. NTG caused hyperalgesia, upregulated CGRP…”

  1. Please present more articles, preferably more than 150 for review articles. I suggest the authors focus their efforts on researching relevant literature: I believe that adding more citations will help to provide a better and more accurate background to this study.

For eaxmple:

https://pubmed.ncbi.nlm.nih.gov/35052756/

https://pubmed.ncbi.nlm.nih.gov/25219387/

https://pubmed.ncbi.nlm.nih.gov/34635045/

Answer: We have added several new references or repeated already cited ones in fragments related to the same reference. However, we did not repeat a reference after each sentence in a sentence-by-sentence logical chain related to it. Moreover, we think that the number of references should be adequate to the content of the manuscript, especially when review articles are cited. That is why we do not think that 150 or any other number of references should be automatically applied to a review article independent of its content and volume. Finally, our manuscript is not about general aspects of migraine, but rathe about its epigenetic characteristics in the context of CGRP action. That is why we cannot develop general aspects of migraine, as they can be easily found elsewhere, and we have tried to show some new perspectives. We cannot write a focused review with too many side motifs and too large general background.

Comment: 4. It would be worthwhile to make a table of the results so far on the epigenetic aspects of CGRP and migraine.

Answer: We have inserted such a table (Table 1) at the end of section 4.3.3.

Comment: 5.     Formal errors:

In many places, the abbreviated form appears in front and the full name is in parentheses (e.g.: calcitonin gene-related peptide (CGRP); calcitonin-like receptor (CLR) etc.) while elsewhere they are reversed. I think the full form should appear first and the abbreviated form should be put in parentheses.  Please, correct these.

For example:

Line 67 CALCA (calcitonin related polypeptide alpha) vs. calcitonin related polypeptide alpha (CALCA)

Line 71 CALCB (calcitonin related polypeptide beta)

Line 77, 92 HLH (helix-loop-helix)

Line 94 FOXA2 (forkhead box A2)

Line 240 SLC2A9 (solute carrier family 2 member 9)

Line 241 DGKG (diacylglycerol kinase gamma)

Line 242 KIF26A (kinesin family member 26A), DOCK6 (dedicator of cytokinesis 6), and CFD (complement factor D)

Line 376 EZH2 (enhancer of zeste 2 polycomb repressive complex 2 subunit)

Line 382, 392 ChIP (chromatin immunoprecipitation)

Line 417 CX3CR1 (C-X3-C Motif Chemokine Receptor 1)

Line 418 IL1B (interleukin 1 beta)

Line 492 SIRT1 (sirtuin 1)

Line 495 COX2 (cytochrome c oxidase 2)

Line 496 PGE2 (prostaglandin E2)

Line 501 MAPK (mitogen-activated protein kinase)

Line 506 c-FOS (fos proto-oncogene)

Line 511 TNF-α (tumor necrosis factor alpha), MPO (myeloperoxidase)

Line 558 TGF (transforming growth factor)

Line 621 SF1 (splicing factor 1)

  • The abbreviated form appears earlier, the full form shows up only later. The full form of the abbreviation must be written at the first mention and the abbreviated form must be enclosed in parentheses. Please correct these.

For example:

Line 89 cAMP – Line 108 cyclic adenosine monophosphate (cAMP)

  • The abbreviation has previously been introduced. After the first appearance, only the abbreviated form should be used.

For example:

Line 68 calcitonin (CT) – Line 135, 137 calcitonin

Line 493 complete Freund's adjuvant (CFA) – Line 500 complete Freund’s adjuvant

  • The abbreviation should appear in parentheses at the first appearance of the full form, not later.

For example:

Line 136 katacalcin – Line 137 CCP-I (katacalcin)

  • The term below is abbreviated twice.

Line 382 and 392 ChIP (chromatin immunoprecipitation)

Line 501 and 509 MAPK (mitogen-activated protein kinase)

  • CALCB is written in italics in Line 71, while not elsewhere (e.g.: Line 153, 158).
  • Line 70 ẞ- CGRP (CGRP-2) vs Line 153 CGRP beta (CGRP-2) – please unify this.
  • The full form of CNS (Line 180) does not appear anywhere in the text.
  • Line 199 double „suggested”
  • There are som typing errors, please check and correct them.

For example:

Line 367 plying

Answer: We have followed all these suggestions.